# Different patterns of clonal evolution among different sarcoma subtypes followed for up to 25 years

Jakob Hofvander [1], Björn Viklund[2], Anders Isaksson [2], Otte Brosjö[3], Fredrik Vult von Steyern[4], Pehr Rissler[5], Nils Mandahl[1] & Fredrik Mertens [1,5]

To compare clonal evolution in tumors arising through different mechanisms, we selected three types of sarcoma—amplicon-driven well-differentiated liposarcoma (WDLS), gene fusion-driven myxoid liposarcoma (MLS), and sarcomas with complex genomes (CXS)—and assessed the dynamics of chromosome and nucleotide level mutations by cytogenetics, SNP array analysis and whole-exome sequencing. Here we show that the extensive single-cell variation in WDLS has minor impact on clonal key amplicons in chromosome 12. In addition, only a few of the single nucleotide variants in WDLS were present in more than one lesion, suggesting that such mutations are of little significance in tumor development. MLS displays few mutations other than the *FUS-DDIT3* fusion, and the primary tumor is genetically sometimes much more complex than its relapses, whereas CXS in general shows a gradual increase of both nucleotide- and chromosome–level mutations, similar to what has been described in carcinomas.

[1] Division of Clinical Genetics, Department of Laboratory Medicine, Lund University, SE-221 84 Lund, Sweden. [2] Science for Life Laboratory, Department of Medical Sciences, Uppsala University, SE-751 23 Uppsala, Sweden. [3] Department of Orthopedics, Karolinska Hospital, SE-171 76 Stockholm, Sweden. [4] Department of Orthopedics, Clinical Sciences, Lund University and Skåne University Hospital, SE-221 85 Lund, Sweden. [5] Department of Clinical Genetics and Pathology, University and Regional Laboratories Region Skåne, SE-221 85 Lund, Sweden. Correspondence and requests for materials should be addressed to J.H. (email: jakob.hofvander@med.lu.se)

Genetic instability is considered an obligate feature of cancer cells[1–3]. This assumption is based on theoretical considerations as well as on extensive observations in tumors and experimental systems. Neoplastic transformation is thought to require more mutations than can be expected to arise from "normal" mutation rates and neoplasms consistently harbor, often numerous, somatic mutations. Furthermore, many neoplasms show extensive intratumoral heterogeneity with regard to mutations and clonal evolution is frequently observed in tumors that are repeatedly sampled during disease progression[4–8]. However, most of the conclusions have been drawn from data on highly malignant epithelial neoplasms in adults, which may develop through mechanisms that differ from other solid tumors or hematopoietic malignancies, and data on neoplasms that have been followed for many years are scarce. Finally, while it is a well-established fact that different tumor types show different mutational profiles and that nucleotide level mutations predominate over chromosomal rearrangements in some tumors and vice versa in others[9,10], it remains poorly investigated to what extent these factors affect clonal evolution.

In this context, sarcomas constitute an interesting group of malignancies. Sarcomas are clinically and genetically heterogeneous and can be arbitrarily subdivided into three main subgroups on the basis of their defining genetic characteristics. One subgroup, comprising about 25% of the entities, is characterized by specific gene fusions, which are thought to function as master switches of transcriptional programs; these sarcomas range in clinical behavior from relatively benign to highly malignant[11]. A second subgroup displays supernumerary ring chromosomes, containing amplified material from large genomic segments; these tumors are typically low-grade malignant and display lipoblastic differentiation, but have the, for sarcomas, unusual potential to progress from low-grade to high-grade malignant lesions[12]. The third and largest subgroup shows various and often extensive combinations of genomic imbalances and point mutations, none of which is specific for any given tumor type; these sarcomas are typically medium-high grade malignant[13]. Few studies on genetic instability and clonal evolution have been performed on sarcomas[14–17], and to our knowledge no attempt has been made to compare patterns of clonal evolution in different genetic subgroups.

In order to assess the type and rate of clonal evolution in different pathogenetic subgroups of sarcoma, we selected the two most common subtypes of liposarcoma: well-differentiated liposarcoma (WDLS, aka atypical lipomatous tumor) and myxoid liposarcoma (MLS). WDLS displays supernumerary ring chromosomes containing amplified material from multiple genomic segments, always including substantial portions of chromosome arm 12q[12]. Extensive inter-cellular genetic variation caused by mitotic instability of the ring chromosomes has been demonstrated[18]. MLS is gene fusion-driven—most cases display a *FUS-DDIT3* chimera, which is considered a strong driver mutation[19]. For comparison, sarcomas representing the third genetic subgroup, with complex genomes (CXS), were included. We also studied multiple samples from some of the primary lesions, in order to evaluate intra-lesional heterogeneity. We show that the extensive single-cell variation in WDLS has little impact on key amplicons in chromosome 12, that MLS displays few mutations other than the *FUS-DDIT3* fusion, and that CXS in general shows a gradual increase of both nucleotide- and chromosome–level mutations.

## Results

### Amplicon-driven well-differentiated liposarcomas.
From five patients with WDLS both chromosome and nucleotide level data were available from 20 samples from 12 lesions. Time interval between first and last sampling was 57–306 months (Table 1). All successfully analyzed samples showed composite karyotypes, united by one or more supernumerary ring chromosomes. The inter-cellular variation was extensive: the number and/or size of ring chromosomes varied considerably (Supplementary Fig. 1), and there were numerical as well as structural non-clonal changes; the latter were found in 42% of the cells (Supplementary Table 1). Neither SNP array analyses nor WES reflected this extensive variation (Fig. 1; Supplementary Fig. 2). When comparing three different samples from the same primary tumor (PT) in four cases, no differences were found (Supplementary Data 1). The 12 lesions showed 22–51 (median 35) GCS at SNP array, almost all of which were gains. When comparing any two lesions from the same patient, 25–83% (median 49%) of the breakpoints were shared (Supplementary Data 2), and the median overlap was 0.57 when the total extension of GCS was compared (Supplementary Table 2). The amplified sequences in chromosome 12, including genes, such as *MDM2*, *HMGA2*, and *CDK4*, displayed greater overlap among different lesions from the same patient both with regard to shared breakpoints (range 31%–89%, median 65%; Supplementary Data 2) and to the extension of GCS (range 0.53–0.99, median 0.71; Fig. 2; Supplementary Table 2). Each WDLS sample had few ESV (range 1–11, median 7), at low allele frequencies (median 21%). Intra-lesional heterogeneity was low, with 82–100% of ESVs being present in all three samples analyzed. With time, however, most mutations were unique for each lesion: only 3/72 mutations that were detected were shared with another lesion (Supplementary Data 3). At relapse, the number of GCS did not increase, and ESV only moderately so (Table 1), and there was no indication that the samples became less similar with time. Actually, both cases from which three samples could be analyzed showed greater similarity with regard to GCS on chromosome 12 between the first and last samples (0.97 and 0.99, respectively) than between the first and second or second and third samples (0.69–0.72).

### Gene fusion-driven myxoid liposarcomas.
From nine *FUS-DDIT3*-positive MLS, the PT and 1–4 local recurrences (LR) and/or metastases (Met), occurring 12–104 months after diagnosis, were studied (Table 1). The inter-cellular variation at G-banding was small: among the 317 cells from the 15 samples that could be assessed, only 4 cells (1.3%) showed non-clonal structural aberrations and 4 (1.3%) deviated from the stemline chromosome number (Supplementary Table 1); clonal karyotypes were consistently identical when comparing 2–3 samples from the same PT (Supplementary Data 4). Combining cytogenetic and SNP array data, 1–6 chromosome level aberrations were found per PT and there were few differences (range 0–8, median 1) between a PT and its LR or Met. Two LR (cases 1 and 6) had fewer chromosome aberrations than their PT and 6/13 Met had the same number of chromosome level aberrations as the PT (Fig. 1; Supplementary Data 2; Supplementary Fig. 2). WES data on 11 samples from four patients showed 7–165 (median 15.5) ESV per PT. In MLS 1–3, most (61–100%) ESV detected in a PT were present also at relapse, but Case 4 showed a dramatic decrease. That PT had 165 ESV; the large number of ESV was confirmed at independent WES and targeted re-sequencing. Its four Met, occurring 19–74 months after diagnosis, had only 11–24 ESV. However, the clonal relationship between the PT and the Met was unquestionable, with six ESV being shared by all samples. In addition, they all shared the six chromosome level aberrations seen in the PT, with only 1–2 new aberrations per Met (Fig. 1; Supplementary Tables 3 and 4; Supplementary Fig. 2).

### Sarcomas with complex genomic aberrations.
For comparison with gene fusion- and amplicon-driven liposarcomas, we

**Table 1 Longitudinal genetic study of three different pathogenetic subgroups of sarcoma**

| Case no.[a] | Material[b] | Dx[c] | ESV[d] | GCS[e] | G-band[f] |
|---|---|---|---|---|---|
| 1A | PT | MLS | 7 | 4 | 5 |
| 1B | LR2 (22) | | 9 | 4 | 4 |
| 2A | PT | MLS | 18 | 0 | 1 |
| 2B | Met1 (30) | | 20 | 0 | 1 |
| 2C | Met2 (92) | | ND | ND | 2 |
| 2D | Met3 (98) | | ND | ND | 2 |
| 3A | PT | MLS | 12 | 0 | 1 |
| 3B | Met1 (19) | | 23 | 8 | 1 |
| 3C | Met2a (28) | | ND | ND | 1 |
| 3D | Met2b (28) | | ND | ND | 1 |
| 3E | Met3 (32) | | ND | ND | 1 |
| 4A | PT | MLS | 165 | 2 | 5 |
| 4B | Met1 (19) | | 11 | 3 | 5 |
| 4C | Met2a (20) | | 16 | 3 | 5 |
| 4D | Met2b (20) | | 14 | 3 | 5 |
| 4E | Met3 (74) | | 24 | 4 | 5 |
| 5A | PT | MLS | ND | 0 | 1 |
| 5B | LR1 (104) | | ND | 1 | 1 |
| 6A | PT | MLS | ND | 5 | 3 |
| 6B | LR1 (12) | | ND | 0 | 1 |
| 7A | PT | MLS | ND | 1 | 1 |
| 7B | LR1 (25) | | ND | 1 | 1 |
| 8A | PT | MLS | ND | ND | 4 |
| 8B | Met1 (48) | | ND | ND | 4 |
| 9A | PT | MLS | ND | ND | 1 |
| 9B | Met1 (42) | | ND | ND | 1 |
| 10A | PT | WDLS | 5 | 38 | 8 |
| 10B | LR1 (197) | | 1 | 35 | 3 |
| 10C | LR2 (306) | | 8 | 37 | 2 |
| 11A | PT | WDLS | ND | ND | ND |
| 11B | LR1 (84) | | 4 | 35 | 1 |
| 11C | LR2 (124) | | 7 | 22 | >10 |
| 11D | LR3 (141) | | 9 | 35 | 6 |
| 12A | PT | WDLS | 6 | 38 | 4 |
| 12B | LR1 (124) | | 4 | 51 | >3 |
| 13A | PT | WDLS | 11 | 27 | 11 |
| 13B | LR2 (215) | | 7 | 23 | 4 |
| 14A | PT | WDLS | 7 | 35 | 3 |
| 14B | LR1 (211) | | 8 | 39 | 5 |
| 15A | PT | MFS | 32 | 91 | |
| 15B | LR1 (47) | | 46 | 99 | |
| 15C | Met1 (114) | | 68 | 84 | |
| 16A | PT | MFS | ND | ND | |
| 16B | LR1 (60) | | 29 | 33 | |
| 16C | LR2 (100) | | 31 | 22 | |
| 16D | LR6 (152) | | 33 | 28 | |
| 17A | PT | MFS | 5 | 151 | |
| 17B | Met3 (77) | | 19 | 114 | |
| 18A | PT | MFS | 15 | 27 | |
| 18B | LR2 (110) | | 13 | 30 | |
| 19A | PT | MFS | 25 | 80 | |
| 19B | Met1 (86) | | 27 | 141 | |
| 20A | PT | Myoep | 10 | 110 | |
| 20B | LR2 (294) | | 9 | 99 | |

The highly complex karyotypes in CXS tumors precluded any attempt to calculate the number of aberrations
[a]Multiple samples were analyzed from the primary tumors of cases 5, 7, 10,12–14, and 18–20. The figures for each case denote the combined number of changes in all samples
[b]PT = primary tumor, LR = local recurrence, Met = metastasis. Time in months from diagnosis is indicated in parentheses
[c]Diagnosis. MLS = myxoid liposarcoma, WDLS = well-differentiated liposarcoma, MFS = myxofibrosarcoma, Myoep = myoepithelial tumor
[d]ESV = No. of non-synonymous exonic variants detected at whole-exome sequencing. ND = not done. The values for the PT for which >1 sample was analysed represent the median for all samples
[e]GCS = No. of chromosomal imbalances detected at SNP array analysis
[f]No. of clonal chromosome aberrations detected at G-banding analysis. ND = not done

investigated 6 CXS with 2–3 lesions per case and an interval of 77–294 months between first and last sampling (Table 1). In two cases, 2 or 3 samples from the PT could be analyzed with regard to intra-lesional heterogeneity using both SNP array and WES; in case 18, no differences were seen between the samples, whereas one of the three samples in case 20 had 7 additional imbalances at SNP array analysis (Supplementary Data 1). In most cases the clonal aberrations detected at banding analysis could only be partly resolved and were hence not sufficiently informative for comparisons between samples, but three samples from the PT of Case 19 were analyzed cytogenetically, showing extensive variation, including a ploidy shift in one sample, in clonal aberrations (Supplementary Data 4). SNP array analysis identified 22–151 (median 87.5) GCS per sample, and the fraction of shared breakpoints in samples from the same patient was 6–83% (median 42%). The median overlap of GCS was 0.58 (range 0.24–0.93). The number of ESV per sample (median 26, range 5–68) was higher for CXS than for liposarcomas (median 7 in WDLS and 16 in MLS), and in all but one patient there was a steady increase with time (Table 1; Supplementary Data 3).

## Discussion

Studies of genetic variation and its role for clonal evolution in tumor cell populations face several problems. For example, the initial driving force(s) for neoplastic transformation may provide different prerequisites for which routes are available and what is needed to sustain and optimize continued proliferation. In the present study, we evaluated the type and degree of stemline variation in multiple lesions from liposarcomas—amplicon-driven WDLS and gene fusion-driven MLS—and other sarcomas characterized by complex genomic rearrangements (CXS) that had been followed for long time periods. Apart from this long-itudinal aspect of clonal heterogeneity, we could study intra-lesional heterogeneity at the genome and nucleotide levels in four WDLS and two CXS, as well as inter-cellular (single cell) variation at the chromosome level in all WDLS and 15 MLS lesions. A caveat of the present study is, of course, that the patients were selected on the basis of having late relapses, and it cannot be excluded that rapidly relapsing sarcomas would have yielded different results. Still, the cohort that was analyzed constitutes a rare selection of solid tumors followed for exceptional time periods, and the data provide some interesting clues to the long-itudinal clonal dynamics in sarcomas.

WDLS, driven by amplification of parts of chromosome 12, with MDM2, CDK4, and HMGA2 as the most important targets[12], displays great inter-cellular variation at the chromosome level, as shown in the present study (Supplementary Table 1). In spite of follow-up periods for up to 25 years, this variation had, however, a minor impact on the composition of the tumor stemlines. Furthermore, the small number of mutations, the low allele frequencies, the small number of shared mutations among different lesions from the same patient, and the absence of mutations shared by different patients all strongly imply that ESV are of little or no significance in WDLS development. Of the 70 genes that displayed mutations, only 8 are included in COSMIC's Cancer Gene Census (https://cancer.sanger.ac.uk/census), and none of these mutations has been reported before in soft tissue tumors. The low frequency or absence of ESV that were shared by all lesions from a patient also suggests that WDLS either develop early in life or that the progenitor cell has undergone far fewer cell divisions before neoplastic transformation than a typical pre-cursor cell in a carcinoma. Clonal dynamics in WDLS instead concern larger copy number changes. The results of the present study show that the genotype in WDLS fluctuates around a set of core amplicons in chromosome 12. The only region amplified in

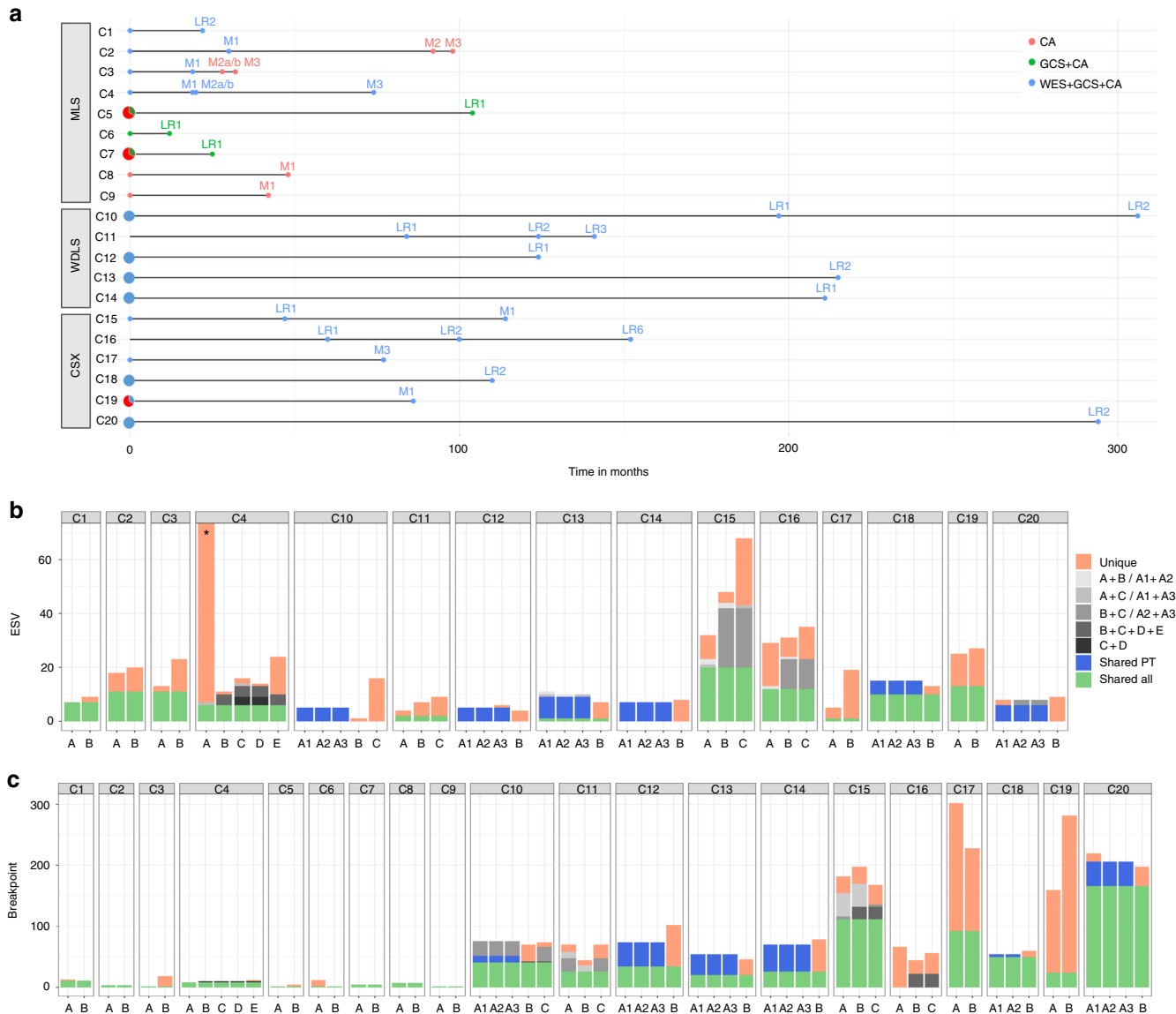

**Fig. 1** Schematic illustration of clonal evolution in 20 sarcomas (C1–C20). C1–C9 are gene fusion-driven myxoid liposarcomas (MLS), C10–C14 are amplicon-driven well-differentiated liposarcomas (WDLS), and C15–C20 are sarcomas with complex genotypes (CXS). **a** Time intervals (in months) between lesions that were analyzed with regard to chromosomal aberrations and nucleotide level mutations. Each sample is indicated by a filled circle; blue samples were analyzed by whole-exome sequencing (WES), SNP arrays (GCS), and chromosome banding analysis (CA), green samples by GCS and CA, and red samples only by CA; larger filled circles represent lesions from which multiple samples were analyzed for assessment of intratumoral heterogeneity. Each line starts with the primary tumor, followed by local recurrences (LR) and/or metastases (M). **b** Diagram showing the number of non-synonymous exonic variants (ESV) detected at WES, as well as the extent of shared mutations among different samples and lesions from the same patient. **c** Diagram showing the number of clonal chromosomal breakpoints detected at GCS and, for MLS also including CB, as well as the extent of shared aberrations among different samples and lesions from the same patient. Figures for C8 and C9 are based on CB only

all 12 samples was a discontinuous 856 Kb sequence in 12q14–15, including six functional genes (Supplementary Data 5), suggesting that at least some of them, notably *MDM2* and the first three exons of *HMGA2*, are essential for tumorigenesis, in line with previous data[12]. Bearing in mind the mitotic instability of ring chromosomes it is highly surprising not only that the follow-up samples were so similar to the first sample, but also that the total extension of chromosome 12 amplification remained 20 times larger than the minimal shared region of amplification. For instance, in Case 10, the total length of the amplified material from chromosome 12 was 19.1 Mb in the PT, 20.7 Mb in the 19 cm large LR1 16 years later, and 19.6 Mb in the 20 cm LR2 another 9 years later (Supplementary Data 2). These results are in line with the suggestions by Lloyd et al. that tumors, as long as

their microenvironment remains stable, relatively early might reach a genetic fitness maximum[20]; additional mutations occur but are not selected for or even deleterious, and transient clones could be attributed to genetic drift facilitated by the bottlenecks caused by the surgical excisions. Indeed, all WDLS follow-up samples analyzed were LR, arising in the same location as the PT and none of the patients had received any chemotherapy that could have shifted the selection pressure.

In MLS, expression of *FUS/DDIT3* has been shown to be sufficient for neoplastic transformation in various experimental models[18], which is in agreement with cytogenetic and sequencing data showing that there are few recurrent chromosomal imbalances, notably trisomy 8 and idic(7)(p11), or exonic SNVs, none of which is consistent[21,22]; the only frequent secondary mutation

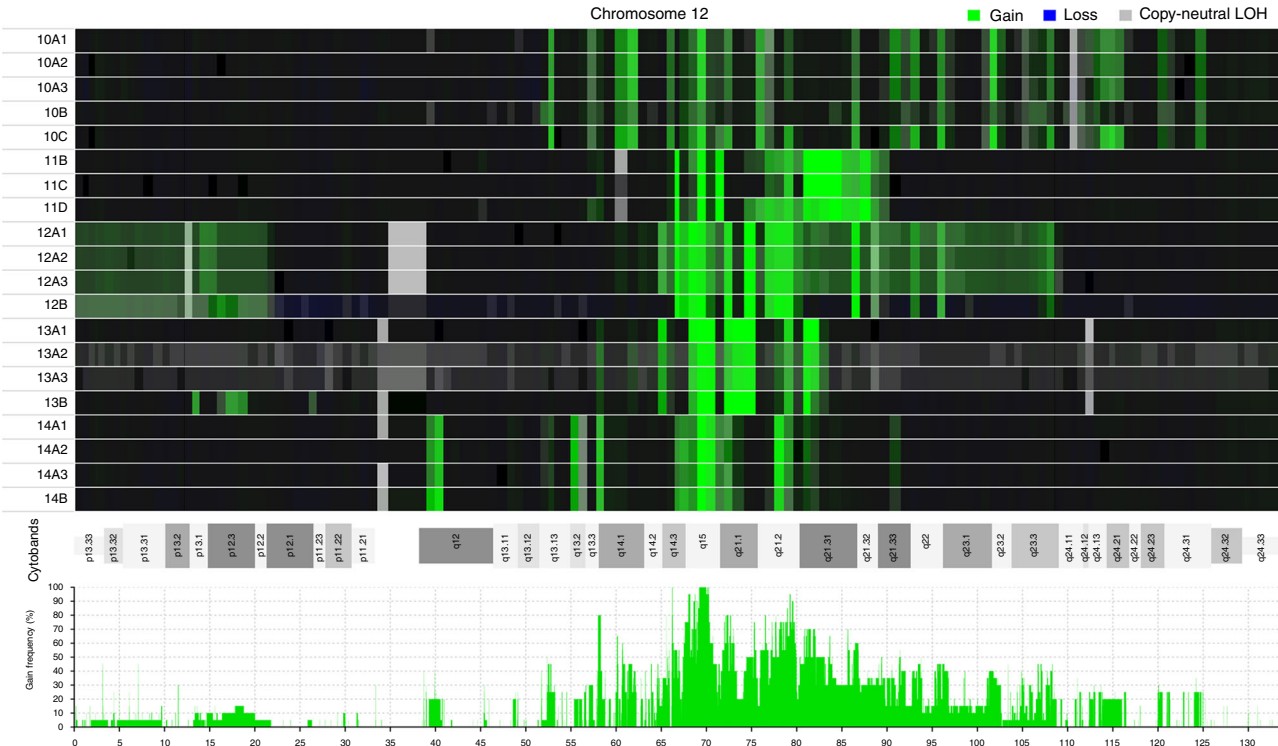

**Fig. 2** Heat map and frequency distribution of amplicons in chromosome 12. Twenty samples from 12 lesions from five patients with well-differentiated liposarcomas, representing amplicon-driven sarcomas, were analyzed. **a** The upper panel, based on the log ratios, shows that the extension of gains (green) and copy-neutral loss of heterozygosity (LOH) is highly similar among different samples from the same patient. Note that samples 10A1–A3 (three samples from primary tumor) and 10C (local recurrence 2; LR2) are more similar to each other than 10A1–3 to 10B or 10B to 10C; the same is true for samples 11B (LR1) and 11D (LR3) in comparison to 11C (LR2). **b** The lower panel, based on the copy number segmentation, shows the frequency of distinct amplicons in chromosome 12 among the 20 samples. Only two segments in 12q14–15, with a combined length of 856 kb, were amplified in all samples

identified so far affects the promoter region of the *TERT* gene, which is seen in some 70–90% of the tumors[23,24]. Despite the relative lack of secondary mutations, the clinical behavior of MLS varies substantially. Some 35% of the patients develop metastases and it has been suggested that certain mutations, e.g., in *PIK3CA* and *TP53*, are associated with aggressive behavior[25–27]. Our results show that clonal evolution in MLS is usually very slow at the chromosome level with few deviations from the stemline, even in metastatic lesions. Less than 5% of the cells showed non-clonal aberrations at G-banding analysis and only 1/4 LR and 7/13 Met showed chromosomal aberrations, as assessed by cytogenetics and/or SNP array, which deviated from the set of shared aberrations (Fig. 1; Supplementary Table 1). Admittedly, 7/13 metastases could only be analyzed by G-banding, but also the six metastases analyzed by high-resolution SNP array showed few (0–8, median 1) additional imbalances compared to the mutational trunk. In contrast, there was a more pronounced accumulation of ESV among the four cases that could be analyzed also by WES, and as expected the relapse samples more often had more ESV than the PT, with the PT of Case 4 as an extreme exception (Fig. 1). That PT had 165 ESV, including in well-known cancer-associated genes such as *BCOR*, *CHEK2*, and *TP53* that have also been implicated in MLS progression[22,25,27]. Only six ESV were shared by all samples, and these occurred at allele frequencies around 5–10% in the PT. Thus, the cell population that gave rise to all metastases had been replaced by a subclone with a much higher level of nucleotide level instability; the allele frequencies of *CHEK2* (54–68%) and *TP53* (36–43%) mutations in this subclone suggest that they occurred early and may have triggered the massive accumulation of ESV.

Case 4 notwithstanding, the results show that MLS cells are genetically relatively stable, and that clonal evolution in MLS is mainly driven by nucleotide level mutations. The slow accrual of new mutations, or even reduction of genetic complexity, in MLS with time and tumor progression has several important implications. First, as already suggested by Reiter et al., cells that eventually form metastases may arise relatively early in the primary tumor; studying pancreatic carcinomas, they showed that metastases share most if not all important driver mutations with their PT[28]. Second, although we cannot exclude an impact of mutations in non-coding sequences, much of the morphological and clinical variation in MLS, such as the transition from a low-grade to a high-grade tumor in cases 3, 6, and 9, could be caused by epigenetic factors. Furthermore, the findings in case 4 demonstrate that therapeutic decisions based on genetic findings in a single sample may not be relevant for all tumor sites. In contrast to the more common notion that mutations in small subclones of a PT might be overlooked, the present case demonstrates that analysis of the PT might suggest therapeutic targets that are not present in the metastatic lesions.

The CXS group of sarcomas was included for comparison with the amplicon-driven WDLS and gene fusion-driven MLS. While the pathogenetic mechanisms in CXS sarcomas still remain relatively poorly investigated, it is well known that there exists an extensive genetic and clinical variation not only among subtypes but also within morphologic subgroups[8,13,29]. In general, our findings in CXS were in good agreement with recent comprehensive genetic data on sarcomas in adults[30]. That study showed that myxofibrosarcomas, which was the most common CXS subtype studied here, have complex copy number changes but few

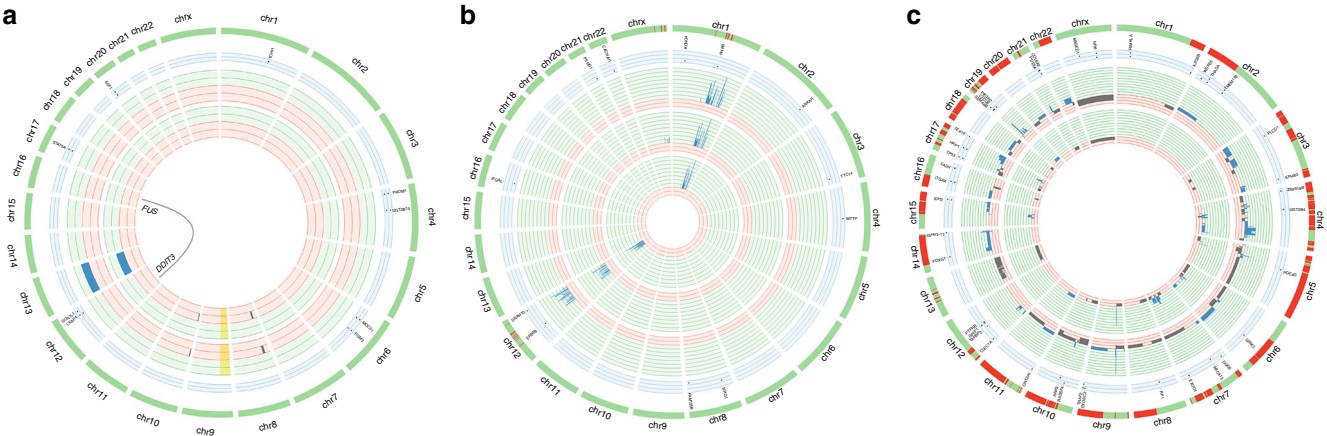

**Fig. 3** Circos plots illustrating different modes of clonal evolution in sarcomas with different genetic backgrounds. **a** A fusion-driven myxoid liposarcoma (MLS), **b** an amplicon-driven well-differentiated liposarcoma (WDLS), and **c** a myxofibrosarcoma (MFS) with a complexly rearranged genome. The red/green inner circles represent the location and amplitude of the allelic imbalances; blue is gain, gray is loss and yellow background indicates loss of heterozygosity. The number of red fields can vary between lesions depending on what is considered the expected number of copies for that lesion in relation to the ploidy level (2n–3n), the number of green fields varies between patients and is determined by the gain with highest number of copies in that patient. The circles are ordered chronologically, starting from the center with the first lesion. The light blue circles represent the location of the variants reported by the whole-exome sequencing (WES) in the same order. Red on the schematic green chromosomes represents differences in genomic changes at SNP array (GCS) between lesions. Both the primary tumor (PT) and the local recurrence (LR) 22 months later from the MLS (case 1) displayed few and mostly identical GCS and ESV. In three lesions from a WDLS (case 10), the GCS of the PT were more similar to those in the second LR occurring after 306 months than to those in the first LR occurring after 197 months. The MFS (case 19) had no less than 209 GCS, but only 39 ESV. While many of the ESV were shared by the PT and the metastasis occurring 86 months later, the GCS overlap was only 0.39. Circos plots for all 20 sarcomas analyzed by both SNP array and WES are shown in Supplementary Fig. 2

significant single nucleotide variants. Indeed, of the 87 mutations that were shared by at least two lesions from the same patient in the present study, only 4 are included in COSMIC's Cancer Gene Census (https://cancer.sanger.ac.uk/census): *EGF*, *IDH2*, *PTPRB*, and *TP53*. Of these, only *TP53* mutations have been implicated in sarcoma development before.

Although, the CXS samples had, on average, higher GCS (median 87.5 compared to 33 in WDLS and 2 in MLS) and ESV (median 26 compared to 7 in WDLS and 16 in MLS) levels than liposarcomas the CXS analyzed here were highly heterogeneous, both with regard to rate and type of clonal evolution. For instance, case 20 showed only 7 ESV in the PT and 11 in the LR obtained 24.5 years later, none of which was shared, but the GCS overlap was 0.79. In contrast, LR1 of case 16 had 29 ESV while LR6 (8 years later) had 35, 12 of which were shared with LR1; at the same time, there were massive changes at the chromosome level, with a GCS overlap of only 0.24 in the two samples (Supplementary Data 2). Thus, more cases of CXS, including other morphologic subtypes than myxofibrosarcoma and myoepithelial tumors, need to be analyzed to draw any firm conclusions on the longitudinal clonal dynamics in these malignancies.

Although it is known that local relapse in sarcoma patients is associated with an increased risk for distant spreading it has been debated whether this should be explained by inherent differences in aggressiveness, i.e., some sarcomas have a higher risk for both local and distant relapse, or whether some locally relapsed tumors actually beget metastases[31]. In the present study, there was only one patient (case 15) with data on both types of relapse: an LR after 47 months and a Met after 114 months. While the PT, LR, and Met all displayed highly complex, incomplete karyotypes, the GCS overlap was higher for the LR-Met comparison (0.73) than for the PT-Met comparison (0.62). In addition, out of 68 ESV occurring at frequencies >5% in the Met, only 1 was uniquely shared with the PT while 20 were uniquely shared with the LR. Thus, the molecular data strongly argue for the LR begetting the Met in this particular case.

In conclusion, the present study shows that the rate by which new mutations become predominant and that the type of clonal evolution, i.e., whether nucleotide or chromosome level mutations prevail, vary considerably among sarcomas caused by different pathogenetic mechanisms (Fig. 3). It also demonstrates, as exemplified by WDLS, that marked genetic instability, i.e., great variation at the single cell level, does not necessarily translate into major changes in the tumor stemline. Whereas, the development of new mutations at the chromosome and nucleotide levels in many CXS fit well with data on carcinomas, both types of liposarcoma displayed a remarkable paucity of clonal evolution at the DNA level. This scenario is similar to what has been suggested for some pediatric tumors and leukemias[32,33], but it should be pointed out that all liposarcoma patients were adults (39–77 yrs). Thus, in some sarcomas the genetic alterations needed for metastatic seeding are present well before the diagnosis of the primary tumor suggesting that they obtain a genetic fitness maximum early in tumor development. As sarcomas are highly heterogeneous from a biological point of view it remains to be investigated whether also other subtypes display similar patterns of clonal evolution. Furthermore, the slow accumulation of DNA level mutations in some sarcomas does not exclude that epigenetic changes could be important in tumor progression.

## Methods

**Tumors.** To assess type and rate of clonal evolution in sarcomas with different pathogenetic mechanisms, we selected patients from which more than one lesion –PT, LR, and/or Met—had been analyzed, and in which at least 1 year had elapsed between first and last sampling. Information on tumors, samples, and analyses performed are given in Table 1 and Fig. 1, and in more detail in Supplementary Data 4. We then combined data from chromosome banding, high-resolution SNP array, and whole-exome sequencing (WES) analyses to assess the spectrum and distribution of genetic aberrations that may develop with time. The study included 20 sarcoma patients from which 2–5 lesions had been obtained with 12–306 months between first and last sampling. Five patients had WDLS, representing amplicon-driven sarcomas, nine had MLS, representing gene fusion-driven sarcomas, and six had myxofibrosarcoma (MFS, n = 5) or myoepithelial tumor (n = 1), representing CXS. Due to the retrospective, longitudinal nature of the

study, with tumors dating back to the early 1980s, only one sample was available from most lesions. However, in nine cases, 2–3 samples from the PT could be studied separately, allowing us to correlate the longitudinal variation with intra-tumoral heterogeneity at the chromosome and/or the nucleotide level. Tumors were diagnosed according to established criteria[29], and the *FUS-DDIT3* fusion transcript in MLS was detected using standard RT-PCR protocols[34]. Gene fusions in Case 20 were excluded through transcriptome sequencing, using previously described methods[35]. Samples were obtained after informed consent and the study was approved by the local review board (diary number 2017/796).

**Chromosome banding and SNP array analysis**. Chromosome preparations were made from short-term cultured cells obtained from disaggregated tumor tissue from 54 samples from 20 patients and stained for G-banding as previously described[36]. SNP array analysis was performed as described[37]. In brief, tumor DNA was extracted from fresh frozen tumor tissue from 54 samples from 43 lesions from 18 patients and analyzed using the Affymetrix CytoScan HD array (Affymetrix, Santa Clara, CA, USA), containing more than 2.6 million markers, or the Illumina HumanOmni1-Quad Genotyping BeadChip (Illumina Inc, San Diego, CA), containing 1.2 million markers. Genomic aberrations were identified by visual inspection using the Chromosome Analysis Suite version 1.2 (Affymetrix) or the GenomeStudio Data Analysis Software (Illumina) combined with bioinformatic analysis regarding copy numbers and segmentation using Rawcopy and the Tumor Aberration Prediction Suite (TAPS)[38,39]. For calculations of intra- and inter-lesional heterogeneity in WDLS and CXS only the genomic changes detected at SNP array analysis (GCS), here including copy-neutral loss of heterozygosity, that extended >500 kb were included. Breakpoints were considered shared when the copy number shift or copy-neutral loss of heterozygosity occurred between the same two probes in two or more samples. For MLS, G-banding and SNP array data were combined to calculate the number of chromosome level aberrations. The human reference sequence used for alignment was the GRCh37/hg19 assembly. Constitutional copy number variations were excluded through comparison with the Database of Genomic Variants (http://projects.tcag.ca/variation/).

**Jaccard index**. The Jaccard index was used to measure the similarity within and between different lesions based on the overlap of their GCS. The index is calculated by taking the ratio of the number of overlapping base pairs between two samples and the length of the union; the union is the length of the GCS in both samples minus the number of overlapping bases. The value of the index can range from 0 to 1, where 0 represents no overlap and 1 represents complete overlap. The Jaccard index was calculated on the genomic intervals listed in Supplementary Table 2 using bedtools (v2.26.0) with the jaccard subcommand.

**Whole-exome sequencing (WES)**. DNA was extracted from fresh frozen tumor biopsies as described[40]. Whole-exome libraries were prepared from a total of 49 tumor samples from 37 lesions and 15 blood samples from 15 patients using the Nextera Rapid Caputre Exome Kit (Illumina) according to the manufacturer's recommendations. Paired $2 \times 76$ bp or $2 \times 151$ bp reads were generated from the exome libraries using a NextSeq 500 (Illumina). First, remaining adapter sequences were removed from the FASTQ files using Trim-galore (v0.4.1). The trimmed reads were aligned to the human reference genome hg19 using BWA-MEM (v0.7.10). Duplicate reads where marked using Picard (v2.2.4) and the BAM files were further processed using GATK (v3.5) according to the best practice pipeline for tumor-normal pairs. Somatic SNVs were called using MuTect[41] (v1.1.7) with default settings and somatic indels were detected using Strelka[42] (v1.0.15) with default settings. Variants were annotated using VEP[43]. The WES generated an average coverage of ×98 of the target bases. The total number of somatic SNVs and indels among all samples were 16,968 and 1428, respectively. In order to enrich for true somatic missense mutations, and limit the number of sequencing artifacts known to be generated by WES, variants were further filtered as follows: read depth of ≥20 in tumor and ≥10 in corresponding normal sample, average base quality ≥20, mutated allele frequency (MAF) of ≥10% in tumor and <1% in the normal sample, and only non-synonymous exonic somatic variants (ESV) were kept. In addition, each variant was visually inspected using IGV (http://software.broadinstitute.org/software/igv/) and a minimum of 2 reads in each orientation was demanded. However, if the same ESV was present in more than one sample from the same patient, only one of the ESV had to fulfill the above criteria and the additional identical ESV needed only 3 reads to be counted. After filtering, a total of 564/861 (unique/total) ESV were retained. An additional "non-somatic" variant caller, Freebayes (v1.0.1) (https://github.com/ekg/freebayes), was run on all samples using the list of filtered variants as targets. This was done for additional verification but primarily to acquire read depth information for positions where no ESV had been reported. The pathogenetic relevance of detected ESV was evaluated with Polyphen (http://genetics.bwh.harvard.edu/pph2/) and SIFT (http://sift.jcvi.org/) and by assessment of COSMIC's Cancer Gene Census database (https://cancer.sanger.ac.uk/census).

**Amplicon sequencing**. In order to verify some of the mutations detected at WES, a TruSeq Custom Amplicon (TSCA) panel (Illumina) was designed. Library preparation was performed according to the manufacturer's recommendations using the TruSeq Custom Amplicon Low Input Kit (Illumina). Paired-end $2 \times 151$ bp reads were generated from the Amplicon libraries using a NextSeq 500 (Illumina). Paired reads were merged using Pear (v0.9.6)[44] and aligned to the human reference genome hg19 using BWA-MEM. SNVs and indels were called on the positions reported from the WES using Freebayes. The TSCA generated an average coverage of ×347 and a total number of 181 variants could be analyzed with sufficient coverage (≥×50). Out of these, 176 (97%) were confirmed and four additional variants missed by the WES were detected.

## Data availability

The data on which the study is based are presented in full in the Supplementary files. The raw data files that support the findings of this study are available from the corresponding author upon reasonable request. Please note that WES data are available for academic purposes by contacting the corresponding author, as the patient consent does not cover depositing data that can be used for large-scale determination of germline variants.

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

## Acknowledgements

The technical assistance of Jenny Nilsson is gratefully acknowledged. This study was supported by the Swedish Cancer Society (CAN2017/269) and Region Skåne (ALFS-KANE-430191).

## Author contributions

J.H., N.M. and F.M. designed research; J.H., B.V., A.I., N.M. and F.M. performed research; O.B., F.V.v.S. and P.R. provided clinical and histopathological data; and all authors assisted with drafting and revising the manuscript.

## Additional information

**Competing interests:** The authors declare no competing interests.

