## [Peer Review File · Nature Communications]

Reviewers' Comments:

Reviewer #1:

Remarks to the Author:

The manuscript by Hofvander et al measures the karyotypes, CNVs, and point mutations in liposarcomas (N= 20 patients). They measured multiple samples from the same patients separated by space and time. The time intervals are relatively long in many cases. Similar studies have been performed in many other cancer types, but this study of sarcomas (rarer than carcinomas) is relatively unique. The numbers of tumors (N=20 patients) are relatively small but sarcomas are rare, and the multiple samples through time adds considerable biological insights into tumor evolution. The overall results are important and interesting.

Some specific comments:

1) The Abstract mentions 2 types of liposarcomas but the paper really studies 3 groups of sarcomas---WDLs (N= 5), MLS (N=9), and CXS (myxofibrosarcoma MFS or myoepithelial tumor, N=6). The problem is continued in Table 1 with "three types of sarcomas". The paper could be clearer on exactly what is being compared.

2) The assays are relatively straightforward and appear to be well-performed. However, the data are somewhat poorly presented. Starting on page 8, the karyotypes vary between cells, which appears to be reflected in Table 1 in the G-band column (?). This reviewer is uncertain how many cells were karyotyped and if the numbers of analyzed metaphases were uniform between samples. The inter-cellular heterogeneity likely reflects chromosomal instability (CIN).

A potential problem is how to present the karyotype data on the trees in Fig 2. Each "Case" is a single branch but the inter-cellular heterogeneity with respect to the karyotypes means each "Case" has multiple subclones. Yet the trees, as drawn, imply single branches. Potentially only the "clonal" changes are used to draw the trees, but the paper should be clearer.

The SNP data are better presented and does not have the subclone problem because only a single sample is studied for each branch (?). The authors should define how they define a shared "breakpoint" (within how many BPs, or does it have to be the exact SNP (?)).

The WES data are interesting. However, the presentation is a bit confusing. Apparently multiple samples in the primary site ("PT") were sequenced but they are summarized by an average in the trees in Fig 2 and Table 1. The heterogeneity between samples in the primary tumor is important information that was obtained, but this information is not well-presented.

3) The data are complex with multiple samples per patient and multiple assays. The data are further complicated because not all assays were performed for all samples. Fig 2 is very difficult to grasp because all of the information is presented on the same tree, and this reviewer struggles to integrate the information. Moreover, even though multiple primary samples were measured, they are not presented, even though this information is really one of the strengths of the paper.

Perhaps separating the data by types in Fig 2 would be easier to comprehend. In addition, tree branch lengths should provide visual information (i.e. be proportional to the units being displayed). One basic tree could be a "time" tree, with branch lengths proportion to the time between specimens---this tree could also clearly illustrate how many specimens were obtained per patient. The next tree could be the karyotype tree, and a CNV tree, and a point mutation tree. I think the point trying to be made with all of these data are that relative to time, the different types of mutations evolve at different rates in the different types of sarcomas. In other words, visually the trees from the same tumor would have different shapes depending on what was measured. The lack of genetic changes with respect to chronological time is striking, but more clarity is desirable to highlight the measured evolutionary tempos at each type of measurement.

This reviewer appreciates how very complex multi-regional and multi-interval genomic cancer data can be. However, clarity of the presentation can increase the impact and understanding of what these relatively unique data disclose.

Reviewer #2:

Remarks to the Author:

The authors reported a study aiming to investigate the impact of genetic heterogeneity on tumor evolution in a subset of sarcomas.

Although, the authors should be commended for having collected a large number of samples, my main concern is that the methodology used do not allow to explore accurately the scientific question.

Indeed, well-differentiated sarcomas and myxoid/round cell liposarcomas are characterized by a very low rate of somatic mutations and copy number alterations (besides 12q amplification in well-differentiated liposarcomas). Therefore, it is difficult to assess temporal heterogeneity without analyzing variability over time during tumor growth and development of the expression profile by using RNAseq. Indeed, heterogeneity does not simply affect coding mutations or copy number alterations. A multitude of epigenetic mechanisms, including DNA methylation, chromatin remodeling, and post-translational modification of histones, can contribute to diversity of expression profile within sarcomas (TCGA Cell 2017). Therefore, claiming that liposarcoma displayed a remarkable paucity of clonal evolution without analyzing the evolution of the expression profile is not valid.

a review, see

Reviewer #3:

Remarks to the Author:

Sarcomas can be loosely grouped into gene-fusion-driven; amplicon (ring chromosome) driven; and ones with complex genetic variation. Here, Hofvander et al chose the two most common liposarcomas:

WDLS (amplicon driven well-differentiated lip sarcoma)

MLS (gene fusion driven myxoid liposarcoma)

as well as CXS (complex karyotype sarcomas) to study clonal evolution and compare and contrast with previous studies in adult solid tumors.

They examine evolution of somatic variants in primary, local recurrence, mets for chromosome banding, high resolution SNP analysis, and WES to identify somatic somatic variants (ESVs). This is performed in 20 patients with 2-5 samples from each, and an overall range from 1-25 years post diagnosis. The cohort includes 5 WDLS, 9 MLS, 6 complex CXS

Methods

The methods are generally standard bioinformatics pipelines that follow best practices, with some additional filtering to remove possible false positives. I have no concerns about the methods.

Main findings

In WDLS there was extensive variation in e.g. number of ring chromosomes from different cells from same WDLS sample - this was not seen in the WES or SNP array analyses. Also no differences were seen in the latter between different samples from the same primary. WDLS samples had few ESVs at low freqs, and they tended to be present in all samples from the same lesion. Overall they were relatively stable with time.

In MLS, few mutations are present other than the driving fusions, and the primary tumor is sometimes more complex than recurrence/mets in terms of mutational burden. Chromosomal aberrations per primary were similar, with few differences between these and recurrences or mets. In most cases, ESVs present in primaries were also in rec/mets but in one case there was a dramatic decrease, though the clonal relationship is noted as questionable.

These are in contrast to results drawn from highly malignant epithelial tumors in adults, where generally speaking there is high mutation burden that tends to increase in late stage tumors such as recurrences and mets.

In the CXS cases there tended to be a steady increase in time of ESVs with variability in the number of shared CNVs.

Strengths:

A unique aspect is length of followup and number of samples per patient. This analysis has not, as far as I am aware, been conducted in sarcomas previously.

Questions and Limitations

The authors argue that low ESV numbers and limited sharing between lesions from the same patient suggest that they are not significant to WDLS development. Not sure I buy this because they could be cooperating lesions with the other variants. In this context, could they comment on the biological relevance of the detected ESVs to sarcomas? There is a missed opportunity to put their findings in the context of the TCGA sarcoma study that was published in December 2017 ([https://www.cell.com/cell/fulltext/S0092-8674\(17\)31203-5](https://www.cell.com/cell/fulltext/S0092-8674(17)31203-5)). Also, these conclusions are based on a small sample size (notwithstanding the uniqueness of the cohort) - how likely are these results to hold up in a larger study?

We would like to thank the reviewers for their critical reading and suggestions for improvements, almost all of which have been possible to take into account; as a result, we believe the manuscript has been strengthened and the message is now more clear. Our response to each of the reviewers' comments are as follows:

Reviewer #1

The manuscript by Hofvander et al measures the karyotypes, CNVs, and point mutations in liposarcomas (N= 20 patients). They measured multiple samples from the same patients separated by space and time. The time intervals are relatively long in many cases. Similar studies have been performed in many other cancer types, but this study of sarcomas (rarer than carcinomas) is relatively unique. The numbers of tumors (N=20 patients) are relatively small but sarcomas are rare, and the multiple samples through time adds considerable biological insights into tumor evolution. The overall results are important and interesting.

Some specific comments:

1) The Abstract mentions 2 types of liposarcomas but the paper really studies 3 groups of sarcomas---WDLs (N= 5), MLS (N=9), and CXS (myxofibrosarcoma MFS or myoepithelial tumor, N=6). The problem is continued in Table 1 with "three types of sarcomas". The paper could be clearer on exactly what is being compared.

Response: In response to these comments, as well as to the requirement of the journal stating that the abstract should contain no more than 150 words, we have rewritten and shortened the abstract (now 150 words). It now includes information on all types of sarcoma included in the study. In line with this comment, and to meet the requirement of the journal, we have changed and shortened the title of the manuscript.

2) The assays are relatively straightforward and appear to be well-performed. However, the data are somewhat poorly presented. Starting on page 8, the karyotypes vary between cells, which appears to be reflected in Table 1 in the G-band column (?). This reviewer is uncertain how many cells were karyotyped and if the numbers of analyzed metaphases were uniform between samples. The inter-cellular heterogeneity likely reflects chromosomal instability (CIN).

Response: We have now updated the former Supplementary Table S1 (now Supplementary Table 6) so that the number of cells belonging to each clone is specified. In addition, we have added karyotypes on separate samples from the same primary tumors in cases 5, 7, and 19. To provide better information on the chromosomal instability of liposarcomas, we have updated former Supplementary Table S3 (now Supplementary Table 1), which previously was focused on well-differentiated liposarcomas, with corresponding data on 317 cells from 15 samples of myxoid liposarcoma. The much smaller level of intercellular variation in the latter type of sarcoma has been commented upon in the text (page 9). Such data are not possible to retrieve from the sarcomas with complex karyotypes; their complexity makes it impossible to exactly assess each numerical and structural aberration in individual cells.

A potential problem is how to present the karyotype data on the trees in Fig 2. Each "Case" is a single branch but the inter-cellular heterogeneity with respect to the karyotypes means each "Case" has multiple subclones. Yet the trees, as drawn, imply single branches. Potentially only the "clonal" changes are used to draw the trees, but the paper should be clearer.

Response: We have clarified that only clonal changes are considered in former Fig. 2 (now Fig. 1), which has been totally revised as suggested by the reviewer (see below).

The SNP data are better presented and does not have the subclone problem because only a single sample is studied for each branch (?). The authors should define how they define a shared "breakpoint" (within how many BPs, or does it have to be the exact SNP (?)).

Response: The definition of a shared breakpoint here is when the copy number transition (or copy neutral LOH) in the samples starts or ends between the same two informative probes. We have added this clarification to the text (page 14).

The WES data are interesting. However, the presentation is a bit confusing. Apparently multiple samples in the primary site ("PT") were sequenced but they are summarized by an average in the trees in Fig 2 and Table 1. The heterogeneity between samples in the primary tumor is important information that was obtained, but this information is not well-presented.

Response: We see the reviewer's point and have updated former Fig. 2 (now Fig. 1) to include both chromosome-level (SNP array) and nucleotide-level data on the multiple samples from primary tumors.

3) The data are complex with multiple samples per patient and multiple assays. The data are further complicated because not all assays were performed for all samples. Fig 2 is very difficult to grasp because all of the information is presented on the same tree, and this reviewer struggles to integrate the information. Moreover, even though multiple primary samples were measured, they are not presented, even though this information is really one of the strengths of the paper.

Perhaps separating the data by types in Fig 2 would be easier to comprehend. In addition, tree branch lengths should provide visual information (i.e. be proportional to the units being displayed). One basic tree could be a "time" tree, with branch lengths proportion to the time between specimens---this tree could also clearly illustrate how many specimens were obtained per patient. The next tree could be the karyotype tree, and a CNV tree, and a point mutation tree. I think the point trying to be made with all of these data are that relative to time, the different types of mutations evolve at different rates in the different types of sarcomas. In other words, visually the trees from the same tumor would have different shapes depending on what was measured. The lack of genetic changes with respect to chronological time is striking, but more clarity is desirable to highlight the measured evolutionary tempos at each type of measurement.

This reviewer appreciates how very complex multi-regional and multi-interval genomic cancer data can be. However, clarity of the presentation can increase the impact and understanding of what these relatively unique data disclose.

Response: We thank the reviewer for these thoughtful comments, with which we fully agree. We

have struggled with finding an alternative that takes all dimensions (time, single-cell, clonal, nucleotide-level, and chromosome-level) into account, but have to confess that we find it impossible to represent all dimensions in a two-dimensional illustration. Following the advice of the reviewer, we have focused on representing the time aspect in the new Fig. 1A, clearly depicting the time span between different samples. In Fig. 1B and 1C, we then show the dynamics of clonal changes at the chromosome and nucleotide levels. Some features, such as single-cell variation in individual samples simply cannot be accommodated in such an illustration, but is now readily available in the revised Supplementary Table 1.

In summary, we believe we have been able to meet all suggestions for improvements made by the reviewer. As a result, the data are now presented in a much clearer way.

Reviewer #2

The authors reported a study aiming to investigate the impact of genetic heterogeneity on tumor evolution in a subset of sarcomas.

Although, the authors should be commended for having collected a large number of samples, my main concern is that the methodology used do not allow to explore accurately the scientific question. Indeed, well-differentiated sarcomas and myxoid/round cell liposarcomas are characterized by a very low rate of somatic mutations and copy number alterations (besides 12q amplification in well-differentiated liposarcomas). Therefore, it is difficult to assess temporal heterogeneity without analyzing variability over time during tumor growth and development of the expression profile by using RNAseq. Indeed, heterogeneity does not simply affect coding mutations or copy number alterations. A multitude of epigenetic mechanisms, including DNA methylation, chromatin remodeling, and post-translational modification of histones, can contribute to diversity of expression profile within sarcomas (TCGA Cell 2017). Therefore, claiming that liposarcoma displayed a remarkable paucity of clonal evolution without analyzing the evolution of the expression profile is not valid.

Response: We fully agree with the reviewer that epigenetic changes are important in tumor development. As shown in numerous studies, cancer cells, as any other multicellular organism, are affected by changes in global and/or gene-specific methylation patterns, chromatin configuration, etc. Indeed, also several sarcomas show critical changes in the regulation of gene expression, as for example in the TCGA study alluded to by the reviewer. To give but two sarcoma examples, 70% of malignant peripheral nerve sheath tumors display loss of H3K27me3 expression and the chimeric proteins causing synovial sarcoma are integral parts of SWI/SNF complexes. It could be pointed out, though, that the factors that could have an impact on selection and differentiation of tumor cells do not end at the traditional epigenetic level. Other levels of interest include post-translational protein modification (e.g., glycosylation of extracellular proteins) and host-tumor interactions affected by factors such as the patient's immunological status or nutrition level. These considerations notwithstanding, very little is actually known about the role of changes beyond the DNA level in tumor evolution, although some highly interesting data have been published (e.g., Hao et al., Nat Genet 2016;36:1500-1507). Part of this ignorance is of course that comprehensive transcriptome or proteome analyses have not been available until recently. Another important aspect is that changes in global gene or protein level analyses are much more difficult to assess than DNA-level mutations, which basically constitute binary data (present or absent). Furthermore, gene expression levels vary to a

much larger extent with non-relevant factors such as handling of samples, methods used, etc. Thus, while categorizing groups of tumors into different major classes with respect to various epigenetic profiles is relatively straightforward, it remains to be shown how subtle differences within or between tumor samples should be interpreted: are these due to differences in the micro-environment, to differences in how samples were handled, or to “true” and stable changes in e.g., methylation patterns? In a study like ours, based on samples obtained during a 30-year period, we can be confident in what we see at the DNA level, but would be hesitant to look at variation at a global gene expression level.

The aim of the present study was to compare the patterns of clonal evolution in sarcomas with what is known from studies of more extensively analyzed malignancies, such as carcinomas. As those previous studies are almost exclusively restricted to DNA-level data, we find it unfair to say that our results are “not valid” without expression profiles. We have already pointed out in the text (Discussion, page 14) that we cannot exclude DNA-level mutations occurring outside coding sequences or epigenetic factors (nor can we exclude any of the other potential contributors mentioned above), but to reiterate this obvious objection, we have now added a sentence to the end of the discussion: “Furthermore, the slow accumulation of DNA level mutations in some sarcomas does not exclude that epigenetic changes could be important in tumor progression.” As a final comment, it could be pointed out that the sarcoma examples provided above (MPNST and synovial sarcoma) display their characteristic epigenetic profiles because of DNA-level mutations in genes regulating these processes, and would hence have been detected here. Similarly, also the study by Hao et al. referred to above shows that the DNA-level and epigenetic fluctuations are not independent of each other.

Reviewer #3

Sarcomas can be loosely grouped into gene-fusion-driven; amplicon (ring chromosome) driven; and ones with complex genetic variation. Here, Hofvander et al chose the two most common liposarcomas: WDLS (amplicon driven well-differentiated lip sarcoma) MLS (gene fusion driven myxoid liposarcoma) as well as CXS (complex karyotype sarcomas) to study clonal evolution and compare and contrast with previous studies in adult solid tumors.

They examine evolution of somatic variants in primary, local recurrence, mets for chromosome banding, high resolution SNP analysis, and WES to identify somatic somatic variants (ESVs). This is performed in 20 patients with 2-5 samples from each, and an overall range from 1-25 years post diagnosis. The cohort includes 5 WDLS, 9 MLS, 6 complex CXS

Methods

The methods are generally standard bioinformatics pipelines that follow best practices, with some additional filtering to remove possible false positives. I have no concerns about the methods.

Main findings

In WDLS there was extensive variation in e.g. number of ring chromosomes from different cells from same WDLS sample - this was not seen in the WES or SNP array analyses. Also no differences were seen in the latter between different samples from the same primary. WDLS samples had few ESVs at low freqs, and they tended to be present in all samples from the same lesion. Overall they were relatively stable with time.

In MLS, few mutations are present other than the driving fusions, and the primary tumor is sometimes

more complex than recurrence/mets in terms of mutational burden. Chromosomal aberrations per primary were similar, with few differences between these and recurrences or mets. In most cases, ESVs present in primaries were also in rec/mets but in one case there was a dramatic decrease, though the clonal relationship is noted as questionable.

These are in contrast to results drawn from highly malignant epithelial tumors in adults, where generally speaking there is high mutation burden that tends to increase in late stage tumors such as recurrences and mets.

In the CXS cases there tended to be a steady increase in time of ESVs with variability in the number of shared CNVs.

Strengths:

A unique aspect is length of followup and number of samples per patient. This analysis has not, as far as I am aware, been conducted in sarcomas previously.

Questions and Limitations

The authors argue that low ESV numbers and limited sharing between lesions from the same patient suggest that they are not significant to WDLS development. Not sure I buy this because they could be cooperating lesions with the other variants.

Response: We thank the reviewer for the positive comments. We agree that a low number of ESVs is not an argument in itself against biological significance; there are many neoplasms with few, but recurrent and important single nucleotide variants. Still, the finding of few and non-recurrent mutations argue against a biological significance. The most important reason for our statement is, however, that very few of the detected mutations in a tumor were found at relapse. The difference between different tumors from the same WDLS patient (only 3 of 72 mutations were found in more than one tumor from the same patient) is in sharp contrast with the finding that different samples from the same primary tumor share the vast majority (28 of 30) of mutations. Thus, even if a cooperative effect of ESVs in a tumor could be imagined, it is difficult to see how such an effect could be achieved when exchanging the entire set of mutations in each new relapse. Furthermore, we know from SNP array analyses that the tumor cell content was high in all samples; still, the ESVs that were detected were found at low allele frequencies, strongly suggesting that they are passenger mutations accruing during tumor growth. Finally, 30/72 ESVs detected in WDLS were classified as benign/tolerated by one of the two predictors (Polyphen and SIFT) that were used. Actually, of the mere 3 ESVs that were preserved with time (i.e., they were present in more than one tumor from the same patient), 2 were classified as benign/tolerated. Allele frequencies and predicted effect of the mutations are available in Supplementary Table 5.

In this context, could they comment on the biological relevance of the detected ESVs to sarcomas ? There is a missed opportunity to put their findings in the context of the TCGA sarcoma study that was published in December 2017 ([https://www.cell.com/cell/fulltext/S0092-8674\(17\)31203-5](https://www.cell.com/cell/fulltext/S0092-8674(17)31203-5)).

Response: We agree with the reviewer that it is of interest to comment on the ESVs detected and to compare the data with the findings in the TCGA study, which is now referred to in the text. In particular, their results on myxofibrosarcomas are highly relevant (myxoid liposarcomas and well-differentiated liposarcomas were not part of the TCGA study). We have assessed the potential impact of the detected ESVs and added comments to the discussion concerning the ESVs in well-differentiated liposarcomas and complex sarcomas. Already in the prior version, we have a discussion on the relevance of the detected mutations in myxoid liposarcomas; apart

from the fact that there are no commonly mutated genes, we emphasize that our results show that a primary tumor can sometimes be much more complex than its relapses, including mutations in well-known cancer-associated genes.

Also, these conclusions are based on a small sample size (not withstanding the uniqueness of the cohort) - how likely are these results to hold up in a larger study?

Response: This question is probably applicable to most studies. Although we believe that the salient conclusions will remain valid also in larger cohorts, we do identify some aspects that should be kept in mind. First, (Discussion, first paragraph) we note that “A caveat of the present study is, of course, that the patients were selected on the basis of having late relapses, and it cannot be excluded that rapidly relapsing sarcomas would have yielded different results”. Second, bearing in mind that the subgroup of sarcomas with complex genomes is a highly heterogeneous one, we have added the following sentence in the discussion (page 12): “As sarcomas are highly heterogeneous from a biological point of view it remains to be investigated whether also other subtypes display similar patterns of clonal evolution.”

In summary, we have revised the manuscript according to all suggestions raised by reviewer 3.

Reviewers' Comments:

Reviewer #1:

Remarks to the Author:

The authors have made the requested revisions. Although the data presentation is clearer, the data are still hard to visualize, albeit the data are complex.

Reviewer #2:

Remarks to the Author:

Revised manuscript suitable for publication since the majority of reviewer comments have been addressed. I am still convinced that clonal evolution in sarcomas (particularly in non-complex genomics ones) is associated with molecular alterations beyond DNA level.

Reviewer #3:

Remarks to the Author:

The authors have addressed my comments.

Response to referees.

No issues were raised by the referees (see below) and we are pleased that they found the revised version of our manuscript suitable for publication.

REVIEWERS' COMMENTS:

Reviewer #1 (Remarks to the Author):

The authors have made the requested revisions. Although the data presentation is clearer, the data are still hard to visualize, albeit the data are complex.

Reviewer #2 (Remarks to the Author):

Revised manuscript suitable for publication since the majority of reviewer comments have been addressed. I am still convinced that clonal evolution in sarcomas (particularly in non-complex genomics ones) is associated with molecular alterations beyond DNA level.

Reviewer #3 (Remarks to the Author):

The authors have addressed my comments.